# STABLE OPTIMIZATION OF GAUSSIAN LIKELIHOODS

## ABSTRACT

Uncertainty-aware modeling has emerged as a key component in modern machine learning frameworks. The de-facto standard approach adopts heteroscedastic Gaussian distributions and minimizes the negative log-likelihood (NLL) under observed data. However, optimizing this objective turns out to be surprisingly intricate, and the current state-of-the-art reports several instabilities. This work breaks down the optimization problem, initially focusing on non-contextual settings where convergence can be analyzed analytically. We show that (1) in this learning scheme, the eigenvalues of the predictive covariance define stability in learning, and (2) coupling of gradients and predictions build up errors in both mean and covariance if either is poorly approximated. Building on these insights, we propose *Trustable*, a novel optimizer that overcomes instabilities methodically by combining systematic update restrictions in the form of trust regions with structured, tractable natural gradients. We demonstrate in several challenging experiments that Trustable outperforms current optimizers in regression with neural networks in terms of the NLL, MSE, and further performance metrics. Unlike other optimizers, Trustable yields an improved and more stable fit and can also be applied to multivariate outputs with full covariance matrices.

## 1 INTRODUCTION

Generating models capable of quantifying uncertainty is crucial in modern machine learning applications. These uncertainties arise, amongst others, due to underlying stochastic processes, inherent fuzziness of the data, or partial visibility (Hüllermeier, 2021). Efficient and reliable strategies for predicting uncertainties allows understanding the underlying data better, interpreting how well a model fares, or guiding decisions we make based on a model.

Conventionally, uncertainty is modeled using a probabilistic approach. That is, a model induces – directly or indirectly – a probability distribution describing uncertainty with its parameters. A well-established technique of estimating aleatoric uncertainty, i. e., uncertainty that accounts for inherent randomness in an experiment, predicts distributions via neural networks (NNs). NNs are extensively addressed methods (Seitzer et al., 2022; Detlefsen et al., 2019; Abdar et al., 2021; Hüllermeier, 2021), primarily due to their ability to learn non-linear and complex relationships. Under the assumption of Gaussian distributed targets, the de-facto standard for various use cases, these NNs infer Gaussian mean and covariance conditioned on the respective input context.

A common objective for training such networks is the negative log-likelihood (NLL) objective, which is given by

$$\theta^* = \arg\min_{\theta} \underbrace{-\sum_i \log p_\theta(x_i, y_i),}_{\mathcal{L}_{\mathrm{NLL}}(\theta, \mathcal{D})} \qquad (1)$$

where $\theta$ are the parameters of the NN and $\mathcal{D} := \{x_i, y_i\}_i$ is the training data. Typically, optimization involves computing some form of gradient $\nabla_\theta \mathcal{L}_{\mathrm{NLL}}(\theta, \mathcal{D})$. Yet, these Gaussian NLL gradient updates were shown to be often unstable and hard to use(Guo et al., 2017; Hüllermeier, 2021). This is for example typically the case if a large portion of training data indicates highly certain subregions – gradients can explode (Takahashi et al., 2018) or fit only sub-par (Seitzer et al., 2022).

Hence, implementations across all domains rely on pragmatic techniques (Seitzer et al., 2022; Garnelo et al., 2018; Schulman et al., 2017a) for stabilizing uncertainty output and gradient training for

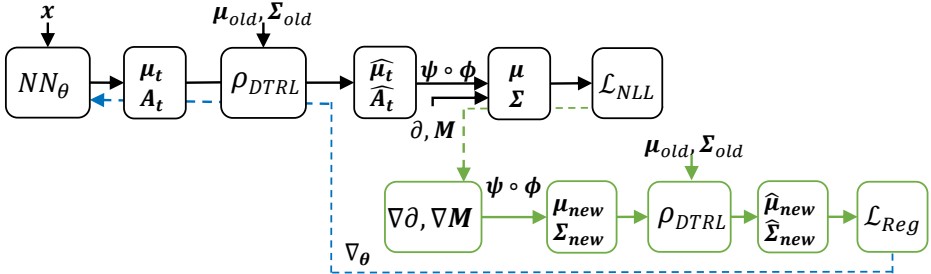

Figure 1: A schematic of Trustable for heteroscedastic prediction of Gaussians. We retrieve the exact natural gradient of the target distributions via local parametrizations (Lin et al., 2021) and regress on the closed-form trust region projected new distributions (Otto et al., 2021). The local loss manifold aware natural gradient cancels out destructive gradient scalings while trust regions stabilize the sensitive update allowing for stable learning of full covariances.

NLL-based learning. Unfortunately, these techniques also impair model expressivity and degrade prediction quality. For example, full covariance matrices in higher dimensions are primarily avoided (Gopinath, 1998; Fiszeder & Orzeszko, 2021) due to their inherent instability during training. A gradient must keep all predicted covariances positive-definite at all times and ensure numerical stability. Most implementations, therefore, do not model correlations but only predict a diagonal Gaussian distribution. Furthermore, clamping the variance into empirically performing regions (Raffin et al., 2021; Garnelo et al., 2018) is a classic makeshift patch to avoid small or large variances in which gradient descent with the Gaussian NLL becomes unstable (Seitzer et al., 2022). In this case, however, the target distribution's true variance cannot be learned if it lies outside the clamped region. Other approaches for compensating destructive gradients include a different variance parametrization (Raffin et al., 2021), objective adjustments (Seitzer et al., 2022) or a change to the optimization strategy (Detlefsen et al., 2019; Choi et al., 2004; Takahashi et al., 2018).

Albeit all these efforts, reliably predicting large ranges of variances and correlations in data remains challenging. The noted shortcomings impact model expressivity and prediction quality. In this work, we examine the underlying cause of those shortcomings and open up new ways to further improve learning for a wide variety of models based on the NLL.

**Summary of Contributions.** We first demonstrate that in an update step with the NLL, the covariance plays a distinct role in the convergence behavior. We extend the idea from Seitzer et al. (2022) and argue that the eigenvalues of the predictive covariance define stability in learning. Furthermore, coupling of gradients and predictions is detrimental as it destabilizes training and can build up errors in both optimization parameters if either is poorly approximated. We then introduce an optimization strategy dubbed *Trustable* (see Figure 1) that methodically overcomes coupling and instabilities. Through structured, tractable natural gradients (NGs) (Lin et al., 2021) we efficiently obtain more informative, decoupled gradients and systematically constrain the update step via Trust Region Projection Layers (TRPLs) (Otto et al., 2021) for stability. Finally, we empirically find that Trustable displays an improved and less volatile fit in terms of the NLL, mean squared error (MSE), and the Wasserstein L2 distance (W2) for non-contextual and contextual regression with full covariance matrices in higher dimensions.

## 2 PRELIMINARIES

In supervised learning, the optimization algorithm receives a labeled set of training data

$$\mathcal{D} := \{x_i, y_i\}_{i \in [0;N]} \subset \mathcal{X} \times \mathcal{Y},$$

where the instance $x_i$ and target $y_i$ are distributed according to a complex, usually intractable distribution $p(x, y)$. Our goal is to approximate the conditional distribution $p(y|x)$ over some data including its aleatoric uncertainty. A common assumption is that each outcome $y$ is distributed according some Gaussian distribution, which is in the multivariate case given by $\mathcal{N}(\mu_x, \Sigma_x)$, where $\mu \in \mathbb{R}^d$ represents the mean and $\Sigma \in \mathbb{R}^{d \times d}_{++}$ the covariance. The covariance is a positive semi-definite matrix

defining the individual variances in its diagonal and correlations in its off-diagonal elements. The eigendecomposition of the covariance matrix is given by $\Sigma = Q^\intercal \Lambda Q$ with $Q$ as eigenvector and $\Lambda$ diagonal eigenvalue matrix. Typically we use strong function approximators such as NNs to map the input values to mean and covariance of the Gaussian distribution, i.e., $f : x \mapsto \mu_x, \Sigma_x$.

For each sample $(x, y)$, the $d$-dimensional multivariate Gaussian defines the NLL via Equation (1) with its density function as

$$\mathcal{L}_{\text{NLL}}^{\text{Gauss}} = -\log p_{\text{Gauss}}(y|\mu_x, \Sigma_x) = \frac{1}{2}\left(d\log 2\pi + \log|\Sigma_x| + (y-\mu_x)^\intercal \Sigma_x^{-1}(y-\mu_x)\right), \quad (2)$$

where $\mu_x \in \mathbb{R}^d$, $\Sigma_x \in \mathbb{R}_{++}^{d\times d}$ are positive-definite. Standard differentiation[1] yields the gradients

$$\nabla_{\mu_x}\mathcal{L}_{\text{NLL}}^{\text{Gauss}} = -\Sigma_x^{-1}(y-\mu_x) \qquad \nabla_{\Sigma_x}\mathcal{L}_{\text{NLL}}^{\text{Gauss}} = -\frac{1}{2}\Sigma_x^{-1}((y-\mu_x)(y-\mu_x)^\intercal - \Sigma_x)\Sigma_x^{-1}. \quad (3)$$

## 2.1 OPTIMIZATION STRATEGIES

Gradient descent is the method of choice for optimizing NNs. Among SGD (Ruder, 2016) and Adam (Kingma & Ba, 2015), we discuss assorted strategies to stabilize and improve gradients for NLL optimization.

**Pitfalls.** Seitzer et al. (2022) indicate that the variance scaling of the gradients introduces importance weights for each sample. As a solution, they propose the univariate $\beta$-NLL loss, multiplying the variance out.

**Trust Regions.** Trust regions are a prominent method in stabilization of non-linear optimization processes (Sun & Yuan, 2006). They produce robust updates by restricting the change in output distributions per iteration which is enforced by bounding the dissimilarity of the new distribution to the old distribution for each update. While many implementations resort to approximations (Schulman et al., 2017a;b), TRPLs (Otto et al., 2021) allow to impose trust regions for each input individually. Formally, the layer solves the constrained optimization problem

$$\underset{\hat{\mu}_x}{\arg\min}\, d_{\text{mean}}(\hat{\mu}_x, \mu(x)), \qquad \text{s.t.} \quad d_{\text{mean}}(\hat{\mu}_x, \mu(x)) \leq \epsilon_\mu \qquad (4)$$

$$\underset{\hat{\Sigma}_x}{\arg\min}\, d_{\text{cov}}(\hat{\Sigma}_x, \Sigma(x)), \qquad \text{s.t.} \quad d_{\text{cov}}(\hat{\Sigma}_x, \Sigma(x)) \leq \epsilon_\Sigma \qquad (5)$$

for each input $x$ to find projected distribution parameters $\mathcal{N}(\hat{\mu}_x, \hat{\Sigma}_x)$ adhering to the trust region. The employed dissimilarity measures are $d_{\text{mean}}$ and $d_{\text{cov}}$, with trust region bounds $\epsilon_\mu$ and $\epsilon_\Sigma$, respectively. For example, we could employ the W2 distance or Kullback-Leibler divergence. The trust region layers can be made fully differentiable due to the use of convex optimization Otto et al. (2021).

**Natural Gradients.** NGs improve convergence by incorporating information about the geometry of the loss manifold through the Fisher information matrix. Instead of expensively computing this matrix per sample (Kakade, 2002; Schulman et al., 2017a), Lin et al. (2021) leverage specific local, auxiliary and global parametrizations on the prior distribution over the parameters. The local parametrization ensures non-singularity of the Fisher, while the auxiliary parametrization connects global and local parametrization via mappings $\psi, \phi$. In local space, the Fisher information matrix collapses into an identity matrix. We employ their mappings introduced for Gaussian distributions

$$\begin{pmatrix}\mu\\\Sigma\end{pmatrix} = \psi(\lambda) := \begin{pmatrix}\mu\\AA^\intercal\end{pmatrix}, \quad \begin{pmatrix}\mu\\A\end{pmatrix} = \phi_{\lambda_t}(\eta) := \begin{pmatrix}\mu_t + A_t\delta\\A_t\exp(\frac{1}{2}M)\end{pmatrix}, \qquad (6)$$

and retrieve the NG for the global space by leveraging the chain rule through the mappings (Lin et al., 2021). This approach alleviates some of the computations from previous NG methods such as computing the FIM. The only overhead is the parameter mapping.

---

[1]The dichotomy for these derivations in various papers is the adhering to additional constraints of symmetric matrix derivatives (specifically $\partial \log|X|/\partial X = X^{-1}$ if $X$ asymmetric vs. $\partial \log|X|/\partial X = 2X^{-1} - X \circ I$ if $X$ symmetric). Indeed, the gradients here are correct; for a detailed discussion of this issue, we refer to Srinivasan & Panda (2020).

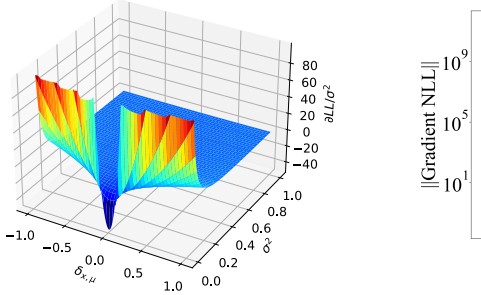

(a) Univariant variance gradient

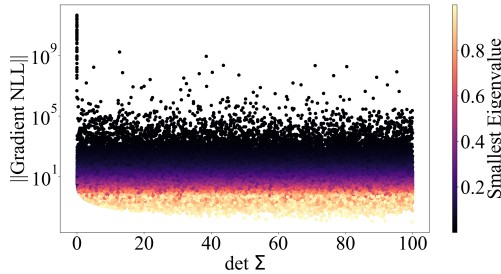

(b) Bivariate gradient magnitudes

Figure 2: The covariance step progression and gradient magnitudes. (a) The univariate variance gradient with a learning rate of $\alpha = 0.01$ on $\delta_{x,\mu} = (x - \mu)$ and $\sigma^2$ experiences various undesirable traits. Values plateau towards larger variances and explode towards smaller ones. The dip in the middle indicates negative (out of parameter space) variance updates. (b) Covariance NLL gradient magnitude on the predictive covariance determinant. Standard Gaussian sampled eigenvalues generate the predictive, bivariate covariance (with randomly rotated eigenvectors). We also sample $\delta_{x,\mu}$ through a standard Gaussian distribution. Overall, small eigenvalues indicate large gradient magnitudes and small $\delta_{x,\mu}$ values indicate high sensitivity to the covariance eigenvalues.

# 3   LIKELIHOOD LEARNING WITH GAUSSIAN DISTRIBUTIONS

## 3.1   ANALYSIS OF NON-CONTEXTUAL GAUSSIANS

While the non-contextual setting allows a closed form solution for mean and variance of a multivariate Gaussian distribution, it is informative to look at the properties of gradient-based optimization already in this setup as we can draw conclusions for the contextual case with NNs as well.

**Gradient Scaling.** Much like Seitzer et al. (2022); Takahashi et al. (2018), we regard the gradient scaling with the covariance as the main culprit for instabilities. However, we want to extend their perspective for univariate Gaussians to the multivariate case. In the univariate case, the gradient scales with $\sigma^{-2}$ for the mean and $\sigma^{-4}$ for the variance. Figure 2a displays this behavior for the univariate NLL gradient around both $\sigma^2 = 0$ and $\delta_{x,\mu} = 0$ and suggests several difficulties. First, cliff-like structures close to $\sigma^2 = 0$ indicate abrupt variance changes, such that even values for a sample error $\delta_{x,\mu} = (x - \mu)$ corresponding to small variances still yield large destructive gradients. In these regions, the gradient is sensitive to noise in $\delta_{x,\mu}$ (cf. zero-variance problem in Takahashi et al. (2018)). This behavior is counter-intuitive, as we would expect progressively smaller gradients, especially when the true variance is small. Second, as the predictive variance $\sigma^2$ becomes larger, a plateau emerges, leading to a slow convergence as large steps are necessary for large variance domains to reach the true variance.

For the multivariate case, we inspect the covariance update through eigendecomposition, which results for a one step with learning rate $\alpha$ in

$$
\begin{aligned}
\Sigma_{\text{next}} &= \Sigma - \alpha \nabla_\Sigma \mathcal{L}_{\text{NLL}}^{\text{Gauss}} \\
&= \frac{2 - \alpha}{2} \Sigma + \frac{\alpha}{2} \Sigma^{-1}(x - \mu)(x - \mu)^\mathsf{T} \Sigma^{-1} \\
\overset{(\Sigma = Q\Lambda Q^\mathsf{T})}{=} &\; Q \left[ \frac{2 - \alpha}{2} \Lambda + \frac{\alpha}{2} \Lambda^{-1} Q^\mathsf{T} \underbrace{(x - \mu)(x - \mu)^\mathsf{T}}_{\text{MC estimate}} Q \Lambda^{-1} \right] Q^\mathsf{T}.
\end{aligned}
$$

The applied gradient interpolates between the current predictive covariance $\Sigma$ and a rescaled version of the Monte-Carlo (MC) estimate $(x - \mu)(x - \mu)^\mathsf{T}$ given the current predictive mean. We now examine the second term from the inside (the inner-most part being the MC estimate) to the outside. Suppose the outer product $(x - \mu)(x - \mu)^\mathsf{T}$ defines an endomorphism. Then, the inner-most transformation $Q^\mathsf{T}(x - \mu)(x - \mu)^\mathsf{T} Q$, a change of basis, expresses the covariance estimate

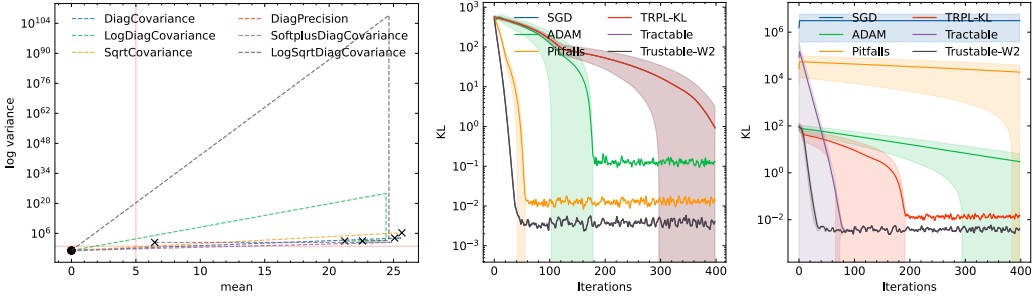

(a) Fitting initially small variance

(b) Fitting initially large (left) and small (right) eigenvalues

Figure 3: Fitting Gaussian distributions with varying parametrizations and optimizers. (a) Training progression of a univariate Gaussian from an initially small variance (●) for 400 iterations (until ×) with SGD. Small variances during training lead to exploding mean and covariance gradients. The true distribution (red lines) cannot be recovered regardless of parametrization. (b) The Kullback-Leibler divergence between true and predictive distribution for a 10D Gaussian with different initial covariance eigenvalues. Standard optimizers struggle, TRPL successfully compensates large, but does not converge for small gradients. While Tractable converges, initial extreme updates can lead to instabilities. In contrast, Trustable shows a fast and stable convergence.

via eigenvectors of the predictive covariance $\Sigma$. After this basis change, the estimate's new space is scaled with the squared inverse eigenvalues of the predictive covariances. In that case, the inverse eigenvalues of the predictive covariance define the stability and convergence behavior along the corresponding eigenvectors. Along eigenvectors with small eigenvalues, large gradients harm the update. Along eigenvectors with large eigenvalues, small gradients slow down convergence. We empirically observe this concept when visualizing the (euclidean) magnitudes of randomly generated two-dimensional covariance gradients in Figure 2b.

**Coupling & Joint Optimization.** The mean and covariance gradients both depend on the predictive mean and covariance. From Equation (3), we can see that the inverse of the covariance matrix scales the mean gradient. Thus, an incorrect covariance influences the mean through eigenvalues and eigenvectors of the matrix and destabilizes training. Vice versa, the predictive mean alters the covariance gradient in Equation (3). To begin with, the MC estimate could be far from the true covariance. This discrepancy arises from a potentially incorrect mean prediction and the stochastically sampled $x$, which allow for possible outliers. Furthermore, the predictive covariance could be arbitrarily wrong, in which case the rescaling is faulty. In other words, if the MC estimate of the covariance for one sample given the predictive mean is far away from the predictive covariance, we end up with significant, potentially destructive changes in the predictive covariance disproportional to the true difference in distributions. This coupling is disadvantageous because it destabilizes the training and can lead to errors in both optimization parameters if either is in a unfavorable region.

**Structural Integrity.** A dip is observed in the middle of Figure 2a as the new predictive variance takes on negative values outside the allowed parameter space. In other words, the NLL gradient itself cannot sustain structural integrity for the predictive variance. Analytically, any function approximator requires to predict a positive-definite matrix in all training iterations. However, the gradient does not ensure positive definiteness for any step. Especially in sensitive regions, the covariance quickly drifts out of the parameter space of positive definite matrices. Numerically, due to the volatility of training for small covariance eigenvalues, a large region of the parameter space must be representable. However, the uneven distribution of floating point numbers and limited representation options lead to fewer actual representable, positive-definite matrices. In particular, for matrices that contain many numbers of sparse float regions (like close to or far away from zero), rounding errors might occur.

**Practical Implications.** For the non-contextual setting, we examine various optimizer and parametrization combinations, progressively increasing the complexity from low- to high-dimensional predictions (see Appendix A.1 and Figure 3). The results confirm theoretical findings and expose eigenvalues and gradient coupling as the main culprits inducing a vicious cycle of harmful, ineffective updates.

Parametrizations can be divided into square-root-based and log-based. The gradient is sensitized by square-root parametrizations, which improves convergence in previously plateau-like areas while exacerbating the small eigenvalue issue. Therefore, the Cholesky parameterization for multivariate predictions experiences instabilities during our experiments. On the other hand, logarithm parametrizations highlight the regions around a covariance of the standard Gauss distribution and improve convergence in the broader range, while degrading gradients further away.

Standard optimizers like SGD or Adam do not compensate for the exploding gradients. The Pitfalls adjustment $\beta$-NLL (Seitzer et al., 2022) with high $\beta$ values canceling out the covariance or update restricting optimizers like TRPL absorb the impact of destructively large gradients. Restricting the update step with trust regions stabilizes the training but slows down convergence, while not compensating for small gradients in regions with large variances. The natural gradient variants converge, but exhibit comparably large gradients in the beginning. Overall, we argue that these gradient explosions harm learning for neural networks. Hence, optimizers that lessen the cycle of damaging updates – by decoupling the gradients, dampening volatile regions, and inciting plateau regions – outperform the competition.

As Figure 3 suggests, none of the above optimization strategies or parametrizations could efficiently mitigate all influences of gradient scaling and coupling. Hence, one must choose an appropriate parametrization $\rho_{\mu,\Sigma}$ to enforce the parameter space integrity, utilize more informative gradients, and restrict the update step for stabilization.

## 3.2 ANALYSIS OF CONTEXTUAL GAUSSIANS

This section extends the above study to the contextual case based on probabilistic NNs. We examine the additional difficulty of adhering to parameter space constraints and sharing model parameters between contexts for prediction.

**Network & Batching Gradients.** Suppose a model defined by $f_\theta : x \mapsto (\mu_x, \Sigma_x)$ is trained via batches. The gradient corresponds to an expectation over various samples

$$\frac{\partial \mathcal{L}_{\text{NLL}}}{\partial \theta} = \mathcal{E}_{(x,y) \sim \mathcal{D}} \left[ \frac{\partial \mathcal{L}_{\text{NLL}}}{\partial \mu_x} \frac{\partial \mu_x}{\partial \rho_{\mu_x}} \frac{\partial \rho_{\mu_x}}{\partial \theta} + \frac{\partial \mathcal{L}_{\text{NLL}}}{\partial \Sigma_x} \frac{\partial \Sigma_x}{\partial \rho_{\Sigma_x}} \frac{\partial \rho_{\Sigma_x}}{\partial \theta} \right], \tag{7}$$

where $\rho_\mu, \rho_\Sigma$ are the parametrizations of the Gaussian mean and covariance, respectively. In our case, both the mean and covariance gradient affect the same parameters as they are shared. If the parameters are shared between the distributions, harmful gradients can spread from each gradient to each distribution parameter. If the size of the network and computational cost allow it, this can be mitigated by splitting the network, i. e., providing a network for the mean and a separate network for the covariance. Similarly, if the parameters are shared between contexts, harmful gradients spread between contexts. This concept of error spreading through the whole network due to a single – or a few – volatile samples destabilizes the training process.

At the same time, batching reduces variance (Bottou, 2012) for the gradient and stabilizes it. For the NLL, it dilates the effect of a subset of harmful gradients. As a result, a NN trains stably as long as a significant portion of the training batch (for all batches) produces meaningful gradients. However, the resulting model is a subpar but stable fit as we either ignore or dampen some training samples.

**Normalization.** Typically, before using a data set for NNs, the data is normalized via mean and covariance. The idea is to center the data and normalize the spread in each dimension. This technique helps to train various types of networks (Bottou, 2012) but does not solve the NLL eigenvalue scaling issue. That is, normalization with global mean and variance does not correspond to normalization of each context individually.

**Subpar Fitting.** As already introduced by Seitzer et al. (2022), when we use batch gradient descent and average over the training batch, each sample defines its own importance (factor) through its gradient magnitude. In a multivariate setting, this weighting results in samples with large gradients, i. e., low predictive covariance eigenvalues, to be responsible for how the learning process is adapting and behaving. Contrarily, samples with small gradients barely influence learning. As a result, suboptimal fits emerge (Seitzer et al., 2022).

Table 1: NLL, MSE, and W2 to the ground truth ($\pm$ standard deviation) for the synthetic datasets.

| | | NNConstant | | | Pitfalls | | | Detlefsen | | |
|---|---|---|---|---|---|---|---|---|---|---|
| Hidden | Model | NLL | MSE | W2 | NLL | MSE | W2 | NLL | MSE | W2 |
| 50x50 | ADAM | $2.93 \pm 1.73$ | $6.54 \pm 0.09$ | $4.93 \pm 0.44$ | $0.09 \pm 0.39$ | $0.08 \pm 0.02$ | $0.0161 \pm 0.0063$ | $3.20 \pm 0.16$ | $38.60 \pm 0.91$ | $1.34 \pm 2.42$ |
| | Pitfalls | $1.80 \pm 0.48$ | $6.41 \pm 0.11$ | $4.90 \pm 0.50$ | $-0.28 \pm 0.42$ | $0.04 \pm 0.01$ | $0.0119 \pm 0.0060$ | $2.90 \pm 0.01$ | $\mathbf{38.19 \pm 0.23}$ | $0.14 \pm 0.08$ |
| | TRPL-W2 | $1.03 \pm 0.20$ | $6.51 \pm 0.05$ | $0.87 \pm 0.43$ | $-0.25 \pm 0.15$ | $0.06 \pm 0.01$ | $0.0209 \pm 0.0098$ | $3.08 \pm 0.20$ | $38.75 \pm 0.45$ | $12.57 \pm 22.14$ |
| | Tractable | $1.52 \pm 0.28$ | $\mathbf{6.19 \pm 0.03}$ | $0.83 \pm 0.39$ | $-0.43 \pm 0.15$ | $0.04 \pm 0.01$ | $\mathbf{0.0039 \pm 0.0015}$ | $2.88 \pm 0.01$ | $38.45 \pm 0.08$ | $0.04 \pm 0.03$ |
| | Trustable-W2 | $\mathbf{0.75 \pm 0.07}$ | $6.21 \pm 0.06$ | $\mathbf{0.17 \pm 0.02}$ | $\mathbf{-0.69 \pm 0.09}$ | $\mathbf{0.03 \pm 0.01}$ | $0.0047 \pm 0.0012$ | $2.90 \pm 0.01$ | $38.64 \pm 0.16$ | $\mathbf{0.03 \pm 0.01}$ |
| 50x50x50 | ADAM | $2.27 \pm 0.40$ | $6.56 \pm 0.05$ | $4.85 \pm 0.47$ | $-0.55 \pm 0.28$ | $0.0348 \pm 0.0186$ | $0.0109 \pm 0.0074$ | $3.81 \pm 2.63$ | $38.58 \pm 0.38$ | $4.06 \pm 10.68$ |
| | Pitfalls | $1.71 \pm 0.67$ | $6.29 \pm 0.05$ | $2.62 \pm 1.29$ | $-0.81 \pm 0.13$ | $0.0170 \pm 0.0079$ | $0.0018 \pm 0.0017$ | $2.90 \pm 0.01$ | $\mathbf{38.31 \pm 0.24}$ | $0.05 \pm 0.01$ |
| | TRPL-W2 | $1.00 \pm 0.18$ | $6.45 \pm 0.11$ | $0.85 \pm 0.20$ | $-0.51 \pm 0.32$ | $0.0392 \pm 0.0231$ | $0.0052 \pm 0.0020$ | $2.92 \pm 0.02$ | $38.88 \pm 0.55$ | $0.09 \pm 0.06$ |
| | Tractable | $2.02 \pm 0.79$ | $\mathbf{6.13 \pm 0.03}$ | $0.56 \pm 0.09$ | $-0.78 \pm 0.15$ | $0.0111 \pm 0.0043$ | $\mathbf{0.0013 \pm 0.0016}$ | $2.90 \pm 0.01$ | $38.39 \pm 0.20$ | $0.04 \pm 0.02$ |
| | Trustable-W2 | $\mathbf{0.66 \pm 0.13}$ | $6.19 \pm 0.04$ | $\mathbf{0.21 \pm 0.06}$ | $\mathbf{-0.94 \pm 0.01}$ | $\mathbf{0.0089 \pm 0.0011}$ | $0.0014 \pm 0.0012$ | $2.91 \pm 0.01$ | $38.55 \pm 0.18$ | $\mathbf{0.03 \pm 0.01}$ |

## 4   TRUST REGION TRACTABLE NATURAL GRADIENT

To recap, we require an optimization strategy that incites low gradient regions, dampens high gradient regions, decouples gradients, and stabilizes training. In this section, we propose *Trustable*, an optimization procedure that aims to satisfy all these. We combine tractable, structured NG (Lin et al., 2021) for improved convergence and a systematic restriction in the form of TRPL (Otto et al., 2021) to stabilize the resulting distributions.

**Non-contextual Trustable Update.** The tractable structured NG update provides an efficient way to retrieve a more informative gradient for the Gaussian NLL loss. It grants information about the local loss manifold to cancel destructive gradient scalings. Additionally, it preserves structural integrity, even with covariance priors or the positive-definiteness constraint of the covariance matrix. The update naturally decouples the mean gradient from the covariance altogether and descales the covariance accordingly (Lin et al., 2021). Since this NG is only a local approximation, we stabilize it by restricting the Gaussian parameter updates via closed-form projection based on TRPLs.

For the forward pass, we first compute the TRPL projected distributions given predictions $(\mu_t, A_t)$. Then, we introduce zero-initialized local parameters $(\delta, M)$ and map them via Equation (6). In the backward pass, we backpropagate through the projections to $(\delta, M)$. According to Lin et al. (2021), the global NG computes through chain-rule to the local parametrization, while the Fisher information matrix reduces to the identity matrix. As a result, the distribution $\mathcal{N}(\mu_{\text{new}}, \Sigma_{\text{new}})$ is the NG adapted version with the computational overhead of evaluating the mappings twice. The tractable update is a local approximation, we must ensure that the distance from new natural parameters to the previous iteration is restricted. Hence, we again project the new natural parameters via TRPL.

We found that this combination improves convergence, as the TRPLs prevent extreme updates of tractable NG while stabilizing the training. In several sample cases (see Appendix A.1) this update scheme converges quicker and provides more stable targets than other tested strategies. Further, across the Gaussian parameter space, the updates keep delivering favorable gradients (see Figure 3b).

**Contextual Trustable Update.** When predicting a non-contextual Gaussian, we have access to its parameters and can update them exactly. For example, the tractable update indeed is the parameters' true NG. Additionally, the model update issue of TRPL (Otto et al., 2021) does not exist; the distribution parameters update directly to the projection. In contrast, for contextual distributions, the Gaussian parameters are defined indirectly via the network. Hence, both the tractable NG and the TRPL cannot be updated by setting the new distributions precisely.

Figure 1 provides a schematic overview for this setting. The NN predicts the mean and cholesky $(\mu_t, A_t)$ of a Gaussian distribution. The TRPL is applied for each output by carrying on old distribution parameters. For each prediction, the TRPL projects its predictive distribution to adhere to a trust region respective the old distribution and the bounds $\epsilon_\mu, \epsilon_\Sigma$, solving Equations (4) and (5). We then introduce zero-initialized local parameters $(\delta, M)$, apply the mappings from Equation (6) to the projected distribution $(\hat{\mu}_t, \hat{A}_t)$ and retrieve our actual predictions $(\mu, \Sigma)$. Note that, in this case, the mappings do not change the projected predictions. They are purely required to compute the NG towards the distribution parameters.

To this end, we compute the NLL loss and backpropagate to the local parameters. As per Lin et al. (2021), the new natural parameters $(\mu_{\text{new}}, \Sigma_{\text{new}})$ can be computed directly through applying the mappings from Equation (6) with $\nabla \delta$ and $\nabla \Sigma$. The non-contextual experiments showed that these new parameters can be extreme (e g. case III of Figure 8). Hence, regressing on them might

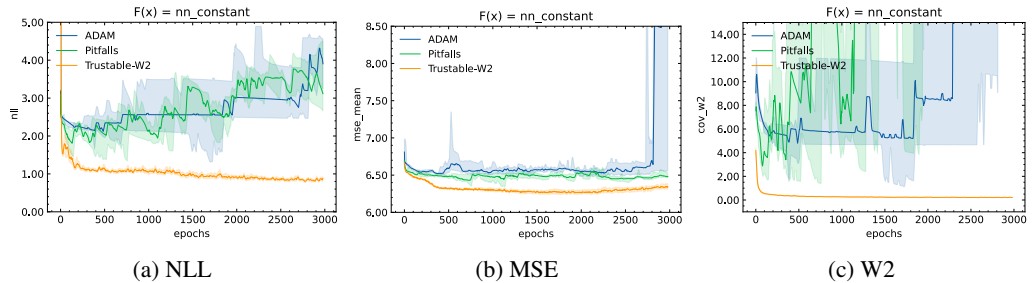

| (a) NLL | (b) MSE | (c) W2 |

Figure 4: Training progression for various optimizers on NNConstant. From left to right, the NLL, MSE between ground truth mean and predictive mean and W2 distance between ground truth covariance and the predictive covariance. The plots represent the median and $85\%$ confidence interval between 10 runs with the best hyperparameters obtained through grid search.

Table 2: UCI regression (multivariate) sets over 10 runs after hyperparameter tuning.

| Optimizer | | NLL | | MSE | |
|---|---|---|---|---|---|
| | | carbon | energy | carbon | energy |
| ADAM | | $-1.18 \pm 1.14$ | $8.00 \pm 0.74$ | $4.43e{-}03 \pm 2.71e{-}03$ | $62.71 \pm 50.11$ |
| Pitfalls | $\beta = 0.5$ | $-12.24 \pm 0.17$ | $7.19 \pm 2.74$ | $\mathbf{9.79e{-}06 \pm 3.11e{-}06}$ | $2.54 \pm 1.18$ |
| Pitfalls | $\beta = 1.0$ | $6.16 \pm 4.61$ | $48.82 \pm 47.75$ | $1.03e{-}05 \pm 7.33e{-}07$ | $\mathbf{0.89 \pm 0.26}$ |
| Trustable-W2 | | $\mathbf{-12.41 \pm 0.19}$ | $\mathbf{3.06 \pm 0.83}$ | $1.30e{-}05 \pm 1.50e{-}06$ | $1.28 \pm 0.69$ |

destabilize our NN. We utilize the TRPL again to retrieve projected targets $(\hat{\mu}_{\text{new}}, \hat{\Sigma}_{\text{new}})$ reasonably close to old distributions and regress on the projected natural parameters to train the network.

For this regression, any distribution metric, such as the W2 or the Kullback-Leibler divergence (KL), can be employed. In our case, we use the same metric that is used in the TRPL to project the distributions.

## 5 EXPERIMENTS

We refer to Appendices A.2 and A.3 for detailed information about datasets, training, and hyperparameters. In the experiments we project via W2 metric, as the non-contextual experiments showed that these deliver good regression targets. Hence, we employ the NLL to evaluate the general model likelihood as well as MSE and W2 to the ground truth for the quality of predictions in terms of mean and covariance.

### 5.1 SYNTHETIC DATASETS

**Step Function with Step Noise (NNConstant).** We first implement a step-wise constant function with segments of varying variances named NNConstant. The challenge in this dataset comes from some segments with minuscule variances, leading to instability or slow learning. Figure 4 presents the confidence over the validation over 10 model runs after hyperparameter optimization. We notice more volatility in the variance prediction in both the NLL and W2 for other optimizers. In contrast, Trustable fits the function and reduces instability in all metrics. We conjecture that through the NG, our optimizer can better trade off the function approximator's allocation to fit the variance better. As a result, the overall likelihood increases. Furthermore, we test for both a two and three layer model (cf. Table 1) and find that only Trustable can fit the data with a smaller model size.

**Sinusoidal without Heteroscedastic Noise (Pitfalls).** We investigate Seitzer et al. (2022)'s illustrative sinusoidal with homoscedastic noise: $y = 0.4 \sin(2\pi x) + \zeta$, with $\zeta \sim \mathcal{N}(0, 0.001)$. As the authors noted, an optimizer requires symmetry breaking to escape locally stable mean fits for such a function. Table 1 display the best models after hyperparameter tuning over 10 trials w.r.t. NLL and W2. While we confirm the results from Pitfalls, where the basic Adam optimizer has issues fitting the function, both the Pitfalls and Trustable converge to a better fit.

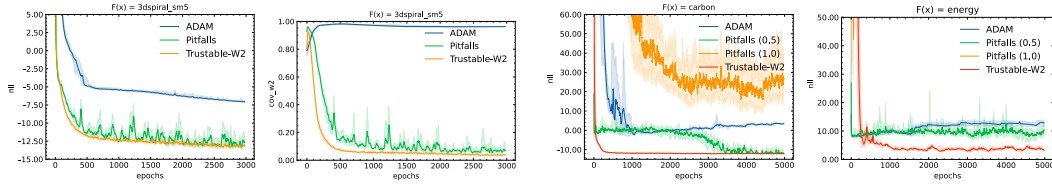

(a) 3DSpiral ($k = 5$), (left) NLL, (right) MSE    (b) UCI Multivariate NLL, (left) carbon, (right) energy

Figure 5: Training progression for various optimizers on multivariate data. (a) NLL and W2 between ground truth and predictive covariance for the 3DSpiral with $k = 5$. (b) NLL on UCI sets *carbon* (3 targets) and *energy* (2 targets) trained with full covariance. All plots represent the median and $85\%$ confidence interval between 10 runs with the best hyperparameters obtained through grid search.

**Sinusoidal with Heteroscedastic Noise (Detlefsen).** The sine from Detlefsen et al. (2019) with additive and increasing noise $y = x\sin(x) + x\zeta_1 + \zeta_2$ with $\zeta_1, \zeta_2 \sim \mathcal{N}(0, 0.3)$ is relatively easy to fit for all methods (see Table 1). Each optimizer learns it reasonably well according to all metrics. We observe that the Trustable optimizer without mean bound converges slightly faster than the Adam and Pitfalls optimizer and, once converged, is stable.

**3D Spiral with Contextual Full Covariance.** This function is based on a three-dimensional spiral of the form $x(t) = r\cos(t), y(t) = r\sin(t), z(t) = at$. We set $r = 0.5$ and $a = 0.05$ to retrieve multiple windings for $t \in [0; 20]$ describing our mean. For the covariance, two eigenvalues are fixed at $0.1$ and $0.2$ and $\mathrm{clip}(-0.01 * (t - 10.0)^2 + 0.1, 10^{-4}, \infty)$ describes the last eigenvalue. Finally, the covariance rotates on the $z$-axis based on $t$. Scaling the covariance with $10^{-k}I$ increases the challenge of learning small eigenvalues. Figure 10 visualizes $k = 0$. We compare Adam, Pitfalls, and Trustable with respect to the NLL and the normalized W2 (Wang et al., 2022) in Figure 5a. Adam does not converge, especially the covariance cannot be fit. Pitfalls cancels out a portion of the destabilizing gradients but remains volatile during optimization in both metrics. Trustable reaches the target distribution in less iterations, typically experiencing less volatility.

## 5.2 UCI REGRESSION DATASETS

We examine the UCI datasets (Hernández-Lobato & Adams, 2015) to check basic convergence. Table 7 presents an overview for Adam, Pitfalls, and Trustable. While Trustable stays on par with other optimizers for the NLL, it improves the MSE on some datasets. On the two datasets with multiple outputs *carbon* (3 targets) and *energy* (2 targets), we learn a full covariance matrix. An overview over best NLL and MSE values are shown in Table 2. Adam does not fit the function properly, achieving worse scores on all metrics. For Pitfalls, convergence depends on $\beta$ but generally improves the fit, but high $\beta$ have a hard time fitting a full covariance. The training progress in Figure 5b suggests that, across training, Trustable performs smoother and fits to a lower NLL.

## 6 DISCUSSION

In this work, we report on the instabilities of NLL optimization and highlight the role of the predictive covariance for stability. Along eigenvectors with corresponding small eigenvalues, large gradients destroy the update. Along eigenvectors with corresponding large eigenvalues, small gradients slow down convergence. Furthermore, we argue that coupling between the predictive parameters and gradients causes poor update steps in both optimization parameters if either one is poorly estimated.

We propose Trustable, combining TRPLs and tractable, structured NGs to methodically tackle these issues. The combination proves to be consistently on par or better than comparable optimizers on various regression tasks while stabilizing full covariance matrix prediction. The main limitation of our method lies in carrying old distribution parameters over for projection of the current predictions. While static datasets can benefit from Trustable directly, dynamic data like in reinforcement learning requires more approximations. In future work, we could address this issue by either regressing on the final network or adding an auxiliary loss to incorporate projection information, like Otto et al. (2021).

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

# A APPENDIX

## A.1 NON-CONTEXTUAL IMPLICATIONS

We investigate the convergence of the NLL gradient for various optimizers and well-established parametric alternatives in a non-contextual setting (see Figure 6). We sample batches of data directly

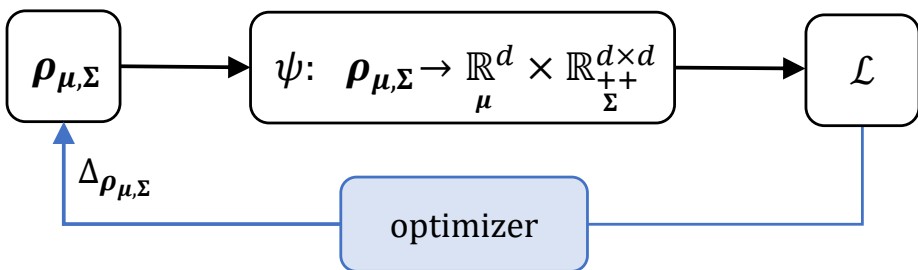

Figure 6: In the non-contextual setting, one Gaussian distribution is predicted and parametrized directly by $\rho_{\mu,\Sigma}$. We sample from the true distribution and compute the NLL loss, adapting the parameters as the respective optimizer defines.

from the true distribution and iteratively update predictive mean and covariance according to the optimizer. Therefore, we employ commonly known optimizers, trust region based optimizers to restrict destructive updates and more informative gradients such as the natural gradient:

- Stochastic gradient descent (SGD)

- Adaptive moment estimation (Adam)

- Pitfalls adjusted SGD: We dub the $\beta$-NLL loss from Seitzer et al. (2022) *Pitfalls*. In contrast to their work, we expand the authors' idea to higher dimensions by rescaling the eigenvalues along each covariance eigenvector, analogously to one dimension, and retrieve the gradients

$$\nabla_\mu \mathcal{L}_{\beta-\text{NLL}} = \Sigma^\beta \nabla_\mu \mathcal{L}_{\text{NLL}}^{\text{Gauss}} \qquad \nabla_\Sigma \mathcal{L}_{\beta-\text{NLL}} = \Sigma^{\beta/2} \nabla_\Sigma \mathcal{L}_{\text{NLL}}^{\text{Gauss}} \Sigma^{\beta/2}. \tag{8}$$

- Natural gradient (NG): For the NG, we estimate the Fisher information matrix (FIM) per batch of training samples (Kakade, 2002) and utilize a Trust Region Policy Optimization (TRPO)-like update (Schulman et al., 2015) without line search

$$\hat{g}_{\text{NG}} = \eta^{-1} F^{-1} \nabla_\theta \mathcal{L}_{\text{NLL}} \qquad \eta^{-1} = \sqrt{\frac{2\delta}{\hat{g}^\intercal F^{-1} \hat{g}}},$$

  with $\delta$ as the trust region bound in terms of the Kullback-Leibler divergence (KL) between the current and updated Gaussian distribution.

- Tractable Structured NG: We optimize via tractable structured NGs (Lin et al., 2021) utilizing a Gauss parametrization with factorized covariance structure (Cholesky or Squareroot). The gradient propagates to local parameters and computes the exact FIM in the local parameter space, updating the distribution accordingly.

- Trust Region Projection Layer (TRPL): In this algorithm dubbed *Trp* in the plots, the TRPLs (Otto et al., 2021) restrict the update step. We save one set of old distributions $(\mu_{\text{old}}, \Sigma_{\text{old}})$ for the projections and evaluate all three metrics Frobenius distance (FD), W2 and KL.

- Combination of Tractable and TRPL: Combining the tractable NG with trust regions might lead to improved convergence. We dub this combination *Trustable* or *TrpTrac* and test the Tractable update with all three trust region projections.

- Gauss-Newton: Gauss-Newton defines our baseline for these experiments. The NLL is not jointly optimizable (Boyd & Vandenberghe, 2004). Hence, we optimize via block-diagonal Gauss-Newton update. To retrieve stable updates, the optimizer solves a least-square optimization problem

$$H\hat{x} = g \quad \text{with} \quad H = \begin{bmatrix} \nabla^2_\mu \mathcal{L}_{\text{NLL}} & 0 \\ 0 & \nabla_\Sigma \text{Vec}[\nabla_\Sigma \mathcal{L}_{\text{NLL}}] \end{bmatrix}, \tag{9}$$

computing the second order gradient $\hat{x} = [\hat{g}_\mu \quad \text{Vec}[\hat{g}_\Sigma]]^\intercal$. We use Tensorflow's built-in least-square solver with $L_2$ regularization of $10^{-7}$.

The following univariate cases (see Table 4) simulate real-world issues that might arise during optimization:

1. Learning minuscule variances – Initially, a standard variance of $\sigma^2_{\text{init}} = 1.0$ must be turned into the rather small true variance of $\sigma^2_{\text{true}} = 0.01$. We would expect proper training at the beginning, and the smaller our variance becomes, the more inconsistent and unstable.

2. Baseline – Training from $\sigma^2_{\text{init}} = 1.0$ to $\sigma^2_{\text{true}} = 1.0$.

3. Unfavorable initialization – Starting at $\sigma^2_{\text{init}} = 0.01$, we expect the training to be unstable with large leaps in variance and mean immediately. The target is $\sigma^2_{\text{true}} = 1.0$.

Another set of multivariate cases reflect various challenges for the covariance (see Table 4). We draw optimal covariances to learn as true covariances and the following initial covariances per case:

1. Optimal eigenvalues – In an optimal optimization problem, all true covariances are centered around eigenvalues of $1$. Hence, we expect rapid and stable convergence.

2. Mostly small eigenvalues – The initial covariance exhibits mostly minuscule eigenvalues. In this case, the learning process should have large gradients and frequent instabilities, usually not converging to the true covariance.

3. Mostly large eigenvalues – Initially, random large eigenvalues in the covariance. When mostly large eigenvalues lead to small gradients, we expect slow convergence, likely not reaching the optimal covariance.

4. "A mix of II and III" – This most difficult case combines II and III.

We draw the covariance eigenvalues randomly from the distributions depicted in Figure 7 and randomly rotate the matrix in each span direction. Tables 3 and 4 summarize hyperparameters.

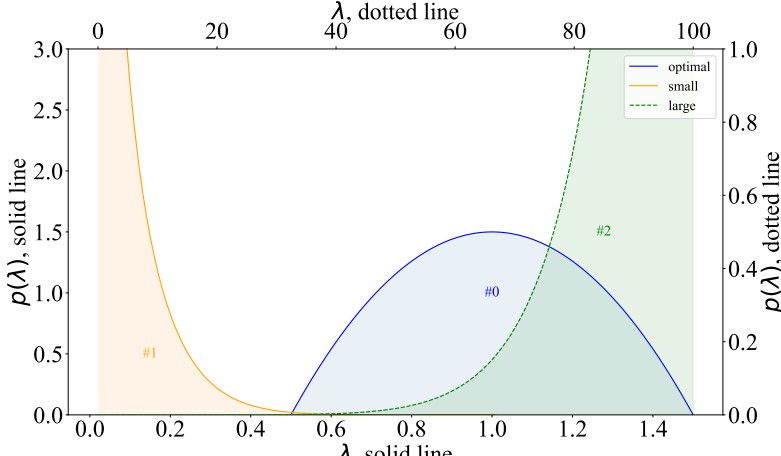

Figure 7: The distribution of eigenvalues per non-contextual, multivariate test case. All case distributions are essentially $\beta$-distributions over the eigenvalues. For Case IV, we Bernoulli sample from # 1 (II) and # 2 (III) with $p = 0.5$.

Table 3: Non-contextual experiments optimizer hyperparameters

| Symbol | Optimizer | Description | Value |
|--------|-----------|-------------|-------|
| | | **Default** | |
| $I$ | * | Number of iterations | 400 |
| $B$ | * | Batch size per update step | 128 |
| $M$ | * | Number of mini batches | 8 |
| $\alpha$ | * | Learning rate | $5e-2$ |
| $I$ | NG | Number of iterations | 50 |
| $\beta$ | Pitfalls | Interpolation coefficient | $\{0.5, 1.0\}$ |
| $I$ | GaussNewton | Number of iterations | 50 |
| | | **Experiments 1D** | |
| $\delta$ | NG | Natural gradient KL trust region | 0.001 |
| $\epsilon_\mu$ | TRPL | Trust region mean bound | 0.1** |
| $\epsilon_\Sigma$ | TRPL | Trust region covariance bound | 0.001** |
| | | **Experiments 10D** | |
| $\delta$ | NG | Natural gradient KL trust region | 0.1 |
| $\epsilon_\mu$ | TRPL | Trust region mean bound | 1.0** |
| $\epsilon_\Sigma$ | TRPL | Trust region covariance bound | 1.0** |
| $\epsilon_\mu$ | TRPL - Force | Trust region mean bound | 0.1** |
| $\epsilon_\Sigma$ | TRPL - Force | Trust region covariance bound | 0.1** |
| $\epsilon_\mu$ | Trustable/TrpTrac | Trust region mean bound | 1.0** |
| $\epsilon_\Sigma$ | Trustable/TrpTrac | Trust region covariance bound | 1.0** |

  * Default parameters are used if not specified otherwise
  ** Bound for all projections Frob/W2/KL

Table 4: Non-contextual experiments cases breakdown

| Case | Description | Initial Distribution | Target Distribution |
|------|-------------|----------------------|---------------------|
| | | **Experiments 1D** | |
| I | Learning minuscule variances | $\mu_{\text{init}} = 0.0$ 
 $\sigma^2_{\text{init}} = 1.0$ | $\mu_{\text{true}} = 5.0$ 
 $\sigma^2_{\text{true}} = 0.1$ |
| II | Baseline | $\mu_{\text{init}} = 0.0$ 
 $\sigma^2_{\text{init}} = 1.0$ | $\mu_{\text{true}} = 5.0$ 
 $\sigma^2_{\text{true}} = 1.0$ |
| III | Unfavorable initialization | $\mu_{\text{init}} = 0.0$ 
 $\sigma^2_{\text{init}} = 0.1$ | $\mu_{\text{true}} = 5.0$ 
 $\sigma^2_{\text{true}} = 1.0$ |
| | | **Experiments 10D\*** | |
| I | Optimal eigenvalues | $\mu_i \sim \mathcal{U}_{[-5.0, 5.0]}$ 
 $\lambda_i \sim \text{Beta}(2, 2) + 0.5$ | $\mu_i \sim \mathcal{U}_{[-5.0, 5.0]}$ 
 $\lambda_i \sim \text{Beta}(2, 2) + 0.5$ |
| II | Mostly small eigenvalues | $\mu_i \sim \mathcal{U}_{[-5.0, 5.0]}$ 
 $\lambda_i \sim \text{Beta}(0.5, 8) + 0.01$ | $\mu_i \sim \mathcal{U}_{[-5.0, 5.0]}$ 
 $\lambda_i \sim \text{Beta}(2, 2) + 0.5$ |
| III | Mostly large eigenvalues | $\mu_i \sim \mathcal{U}_{[-5.0, 5.0]}$ 
 $\lambda_i \sim \text{Beta}(8, 0.5) * 100$ | $\mu_i \sim \mathcal{U}_{[-5.0, 5.0]}$ 
 $\lambda_i \sim \text{Beta}(2, 2) + 0.5$ |
| IV | A mix of II and III | $\mu_i \sim \mathcal{U}_{[-5.0, 5.0]}$ 
 $\lambda_i \sim \text{Beta}(2, 2) + 0.5$ | $\mu_i \sim \mathcal{U}_{[-5.0, 5.0]}$ 
 $\lambda_i \sim \text{Beta}(2, 2) + 0.5$ |

* We sample Eigenvalues according to the distributions
  and compute $\Sigma = Q\Lambda Q^{\intercal}$ for random $Q$

Table 5: M-AUC and mean convergence iteration and time (ITER/TIME) of selected optimizers in the multivariate, non-contextual experiments for 10 runs per algorithm, case and parametrization on random seed.

| | | Case I | | | Case II | | | Case III | | | Case IV | | |
|---|---|---|---|---|---|---|---|---|---|---|---|---|---|
| | | **M-AUC** | **ITER** | **TIME** | **M-AUC** | **ITER** | **TIME** | **M-AUC** | **ITER** | **TIME** | **M-AUC** | **ITER** | **TIME** |
| SGD | Vanilla | 3.14e+0 | 269.70 | 33.94 | 4.48e+3 | - | - | 5.03e+2 | - | - | 4.50e+3 | - | - |
| | Cholesky | 4.08e+1 | 270.20 | 35.59 | 2.69e+6 | - | - | 1.22e+2 | 255.50 | 43.95 | 3.09e+6 | - | - |
| | SqrtCovariance | 4.11e+1 | 270.20 | 35.04 | 2.61e+6 | - | - | 1.22e+2 | 255.50 | 42.80 | 3.03e+6 | - | - |
| Adam | Vanilla | 2.54e+0 | 157.80 | 21.46 | - | - | - | 2.98e+2 | - | - | - | - | - |
| | Cholesky | 8.67e+0 | 109.60 | 15.73 | 1.98e+1 | 91.20 | 52.10 | 9.52e+1 | 151.40 | 21.39 | 2.43e+1 | 70.70 | 54.61 |
| | SqrtCovariance | 6.65e+0 | 117.40 | 16.54 | 1.80e+1 | 100.00 | 52.69 | 1.15e+2 | 162.90 | 22.70 | 2.26e+1 | 71.00 | 54.06 |
| NG | Vanilla | 1.27e+1 | - | - | 7.33e+1 | - | - | 1.93e+1 | - | - | 8.30e+1 | - | - |
| | Cholesky | 1.26e+1 | - | - | 7.33e+1 | - | - | 1.97e+1 | - | - | 8.28e+1 | - | - |
| | SqrtCovariance | 1.26e+1 | - | - | 7.33e+1 | - | - | 1.97e+1 | - | - | 8.28e+1 | - | - |
| PF 0.5 | Vanilla | 1.22e+0 | 112.30 | 23.00 | 5.11e+2 | - | - | 4.61e+2 | - | - | 5.29e+2 | - | - |
| | Cholesky | 4.50e+0 | 41.90 | 9.14 | 2.96e+4 | 73.70 | 71.99 | 1.54e+1 | 48.50 | 10.71 | 3.51e+4 | 79.20 | 78.18 |
| | SqrtCovariance | 4.49e+0 | 41.90 | 9.40 | 2.79e+4 | 71.50 | 74.90 | 1.54e+1 | 48.50 | 10.76 | 3.36e+4 | 115.10 | 75.71 |
| TRP W2 | Vanilla | 2.07e+0 | 184.80 | 56.68 | 2.19e+0 | 49.20 | 15.94 | 5.03e+2 | - | - | 2.53e+0 | 58.50 | 17.38 |
| | Cholesky | 7.79e+0 | 122.30 | 35.65 | 2.80e+0 | 31.80 | 9.53 | 1.22e+2 | 255.50 | 89.67 | 3.24e+0 | 35.50 | 10.32 |
| | SqrtCovariance | 7.95e+0 | 123.10 | 33.21 | 2.79e+0 | 33.80 | 9.77 | 1.22e+2 | 255.50 | 88.95 | 3.17e+0 | 38.30 | 10.97 |
| Tractable | TractableCholesky | 8.88e-1 | 24.60 | 12.52 | - | - | - | **6.80e+0** | **31.70** | 16.11 | - | - | - |
| | ApproximateTractable | **6.56e-1** | 22.80 | **3.86** | 1.67e+3 | 56.80 | 9.24 | **6.80e+0** | **31.70** | **5.27** | 2.32e+3 | 59.70 | 9.62 |
| Trustable-W2 | TractableCholesky | 8.27e-1 | **22.20** | 13.68 | 1.69e+0 | 23.00 | 14.01 | 6.90e+0 | 31.80 | 17.86 | 1.93e+0 | **23.30** | 12.84 |
| | ApproximateTractable | 8.27e-1 | **22.20** | 7.19 | **1.68e+0** | 22.80 | **7.29** | 6.90e+0 | 31.80 | 9.56 | **1.92e+0** | **23.30** | **7.33** |

**Instability.** We visualize the training progress in the univariate case with the KL between true and predicted distribution in Figure 8. For the multivariate case, we evaluate speed and stability for each optimization run. Therefore, we compute the mean area under curve (AUC) over all optimization runs per optimizer and parametrization, as well as time and iterations to convergence (KL $< 10^{-2}$) and obtain Table 5. Runs suffering from numerical instability are marked as (–). The two primary concerns causing instability are: Mean and covariance gradients in each eigenvector direction are scaled with the inverse eigenvalues squared. This scaling results in tiny predictive covariance eigenvalues scaling the gradient up, potentially failing the update owing to huge, non-representable positive-definite or out-of-parameter space matrices. On the other hand, sizeable predictive covariance eigenvalues provide such tiny gradients that there is no convergence. Second, the mean and covariance gradients are linked with the predictions themselves. A poor mean prediction results in a poor covariance gradient, and a poor covariance prediction results in a poor mean gradient. This results in a vicious cycle of harmful updates.

**Parametrizations.** Parametrizations essentially adjust the scaling of the gradients. Two groups crystallize out: Square root parametrizations sensitize the gradient, improving convergence in previously plateau-like areas but increasing the small eigenvalue problem. Logarithm parametrizations emphasize the regions around a covariance of $I$ and improve convergence in the broader range while degrading gradients further away.

**Algorithms.** Converging combinations restrict the gradient directly (trust regions) or indirectly (by rescaling it proportionally to the variance, Pitfalls). Alternatively, they decouple the mean gradient from the variance such that even if one of the two parameters is far off its optimum, the other can still converge (NG, Gauss-Newton, Pitfalls). Canceling out the covariance to some extent is a reoccurring theme: The Pitfalls update does it directly. It turns out that true second-order methods like Gauss-Newton or similar methods like NG also comply with this idea. For example, the Pitfalls mean update with $\beta = 1.0$ is equivalent to the tractable NG mean update (cf. Equation (8) and Equation (12) of Lin et al. (2021)). Furthermore, this update is also equivalent to the Gauss-Newton mean update (cf. Equations (8) and (9)). As a result, we can interpret exponentially smoothed maximum likelihood estimation (MLE) as a second-order optimization for the mean with respect to the NLL.

Unfortunately, none of the investigated optimizer parametrization combos solves all provided test cases efficiently. Simply restricting the gradient through various metrics, as via TRPL, stabilizes learning. Of course, then the tiny gradients are not influenced. On the other hand, the tractable NG update indeed converges even for low eigenvalue cases. However, its intermediate distributions can be extreme, hence unstable.

**Non-contextual Trustable** Finally, the Trustable update combines tractable NGs, which converge albeit with extreme intermediate values, with TRPLs for stabilization. As TRPLs prevent immediate volatility, the distributions never reach extreme values if not supported by the data and do not need to recover from there, which saves time.

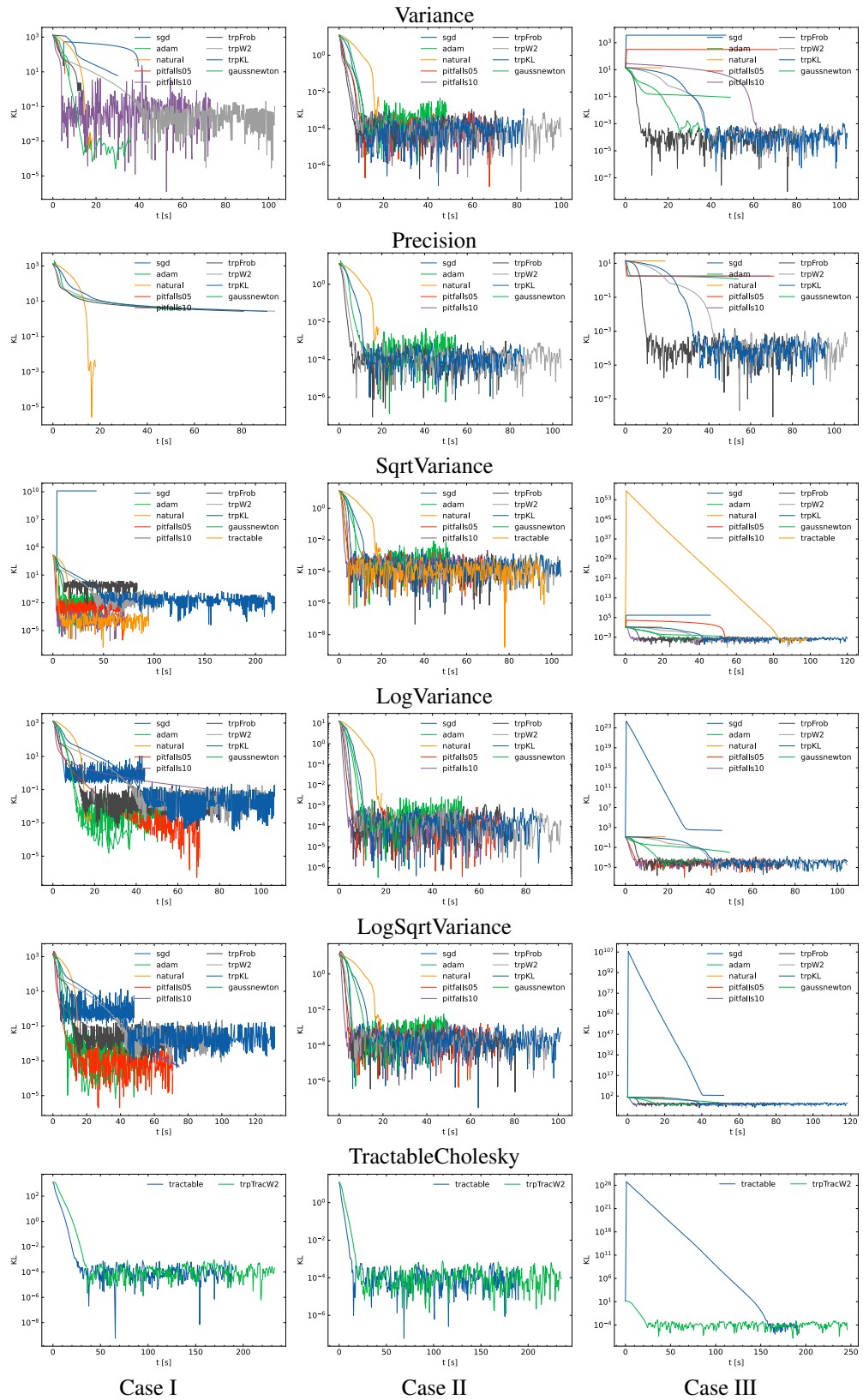

Figure 8: KL between true and predicted distribution for each non-contextual 1D experiment case, parametrization and optimizer on time

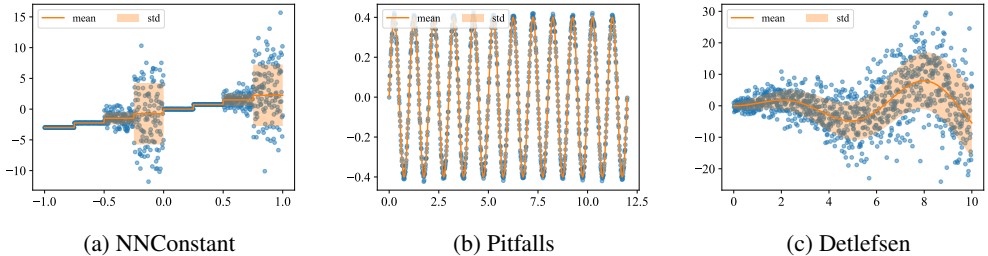

|                    |                   |                    |
|:------------------:|:-----------------:|:------------------:|
| (a) NNConstant     | (b) Pitfalls      | (c) Detlefsen      |

Figure 9: Each plot depicts one function. The ground truth mean and standard deviation are orange, while a typically drawn training set can be observed through the blue samples. All functions map from $\mathbb{R}$ to $\mathbb{R}$ and exhibit contextual variances.

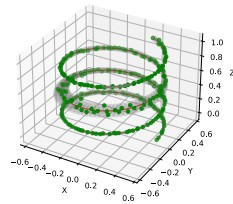

Figure 10: Visualization of mean (red), covariance confidence ellipsoids (grey) and samples (green) of 3DSpiral with $k = 0$.

## A.2 DATASETS AND FUNCTIONS

**Step Function with Step Noise.** This mapping captures complicated to predict discontinuities and in the predictive means and variances through multiple segments. Each segment $i$ can be described by $y = c_{i,1} + \zeta c_{i,2}$ with $\zeta \sim \mathcal{N}(0,1)$. We let $c_{i,1}$ increase equidistantly from $x = [-1.0, 1.0]$ such that the function resides within $y = [-3.0, 3.0]$ and set $c_{:,2} = [0.01, 0.02, 1.0, 5.0, 0.01, 0.02, 1.0, 5.0]$. The result is NNConstant, visualized in Figure 9a. Some segments experience small variances and, thus, influence the whole model through the zero-variance problem and sample importance weightings.

**Sinusoidal without Heteroscedastic Noise.** We utilize the function $y = 0.4 \sin(2\pi x) + \zeta$ with $\zeta$ Gaussian ($\sigma = 0.01$, see Figure 9b) distributed (Seitzer et al., 2022). For training, we evaluate 2000 samples across $x \in [0; 12]$ evenly spaced and keep the variance as ground truth for evaluation.

**Sinusoidal with Heteroscedastic Noise.** We utilize the function $y = x \sin(x) + x\zeta_1 + \zeta_2$ with $\zeta_1, \zeta_2 \sim \mathcal{N}(0, 0.3)$ (see Figure 9c) from Detlefsen et al. (2019). For training, we again evaluate 2000 samples across $x \in [0; 10]$ evenly spaced and keep the function to evaluate the ground truth variance.

**3D Spiral with Contextual Full Covariance.** The function is based on a three-dimensional spiral of the form $x(t) = r \cos(t), y(t) = r \sin(t), z(t) = at$. We set $r = 0.5$ and $a = 0.05$ to retrieve two and a half windings for $t \in [0; 20]$ describing our mean. For the covariance, two eigenvalues are fixed at $0.1$ and $0.2$. The equation $\text{clip}(-0.01 * (t - 10.0)^2 + 0.1, 10^{-4}, \infty)$ describes the last eigenvalue. Finally, the covariance rotates on the $z$-axis based on $t$. To increase difficulty given our analysis stating that small eigenvalues generate learning issues, we scale the covariance by $10^{-k}I$. Figure 10 visualizes the spiral for $k = 0$. We draw 5000 samples across $t \in [0; 20]$ evenly spaced through the respective means and covariances.

**UCI Regression Datasets.** We choose a subset of the UCI datasets employed in Hernández-Lobato & Adams (2015). The data is zero-mean unit-variance normalized on the training set, while the reported NLL is provided in the original scale of the data.

Table 6: Hyperparameters for the contextual experiments

| Symbol | Hyperparameter | Value |
|---|---|---|
| | Default | |
| $\alpha$ | learning rate | $\{1e-4, 5e-4, 1e-3, 5e-3, 1e-2\}$ |
| | batch size | 512 |
| $\beta$ | Pitfalls Beta | 0.5 |
| | Tractable - Regression Metrics | MSE (mean) - W2 (covariance) |
| $\beta$ | Tractable Beta | $\{1e-4, 5e-4, 1e-3, 5e-3, 1e-2\}$ |
| | TRPL - Type | W2 |
| | Detlefsen, Sinusoidals | |
| $\epsilon_\mu, \epsilon_\Sigma$ | TRPL Bounds | mean - $\{1e1, \infty\}$ 
 covariance - $\{1e2, 1e1, 1e0, 1e-1\}$ |
| | 3DSpiral | |
| $\epsilon_\mu, \epsilon_\Sigma$ | TRPL Bounds | mean - $\{\infty\}$ 
 covariance - $\{1e1\}$ |
| | UCI Univariate | |
| | batch size | 256 |
| $\epsilon_\mu, \epsilon_\Sigma$ | TRPL Bounds | mean - $\{1e-1, 1e1, \infty\}$ 
 covariance - $\{1e-4, 1e-3, 1e-1\}$ |
| | UCI Multivariate | |
| $\epsilon_\mu, \epsilon_\Sigma$ | TRPL Bounds | mean - $\{1e-1, 1e1, \infty\}$ 
 covariance - $\{1e-4, 1e-3, 1e-1\}$ |

### A.3 HYPERPARAMETER SETTINGS AND TRAINING

**Step Function, Sinusoidals and 3DSpiral.** We train a multi-layer perceptron with ReLU activations per output parameter, i. e., for mean and covariance separately, predicting the covariance cholesky clamped between $1e-8$ and $1e3$ to be extendable for higher dimensions. For the univariate functions, we use either two hidden layers with 50 neurons each and for the 3DSpiral three hidden layers with 50 neurons each.

We take 70% of the samples as train set and the rest for testing. We split the train set further 70 : 30 and hypertune (over the parameters provided in Table 6) for 1000 epochs with an early stopping patience of 200 monitoring log-likelihood on the validation set. We then train 10 models for 3000 epochs on the train set and evaluate on the test set.

**UCI Regression Datasets.** Like in Hernández-Lobato & Adams (2015), we use the same NN as before but with only one hidden layer with 50 neurons for the univariate case. For the multivariate case we increase to two layers with 50 neurons each.

We follow the same training procedure as for the functions above, but split the data by 20% test and again 20% evaluation. Furthermore, we adjust to 5000 epochs in both hypertuning and on the train set with an early patience of 100 epochs for the univariate and 50 epochs for the multivariate case.

It should be noted that the performance we provide is not comparable to that of other publications, as it is known to vary among data splits.

### A.4 ADDITIONAL RESULTS

**UCI Regression Datasets Univariate.**

Table 7: UCI regression (univariate) sets over 10 runs after hyperparameter tuning.

| | | carbon | concrete | energy | housing | kin8m | naval | power | yacht |
|---|---|---|---|---|---|---|---|---|---|
| | | | | | NLL | | | | |
| ADAM | | $\mathbf{-15.54 \pm 0.12}$ | $\mathbf{3.09 \pm 0.05}$ | $2.93 \pm 0.44$ | $3.16 \pm 0.27$ | $-1.17 \pm 0.03$ | $-13.75 \pm 0.41$ | $2.61 \pm 0.03$ | $\mathbf{1.35 \pm 0.48}$ |
| Pitfalls | $\beta = 0.5$ | $-14.11 \pm 0.69$ | $3.15 \pm 0.07$ | $2.33 \pm 0.28$ | $2.93 \pm 0.12$ | $-1.18 \pm 0.05$ | $\mathbf{-14.49 \pm 0.31}$ | $\mathbf{2.59 \pm 0.03}$ | $1.63 \pm 0.58$ |
| Pitfalls | $\beta = 1.0$ | $-12.05 \pm 0.35$ | $3.17 \pm 0.04$ | $2.85 \pm 0.28$ | $\mathbf{2.90 \pm 0.08}$ | $\mathbf{-1.19 \pm 0.06}$ | $-14.35 \pm 0.28$ | $2.62 \pm 0.03$ | $2.44 \pm 0.45$ |
| Trustable-W2 | | $-11.74 \pm 0.13$ | $\mathbf{3.09 \pm 0.12}$ | $\mathbf{2.32 \pm 0.28}$ | $7.52 \pm 3.16$ | $-1.15 \pm 0.07$ | $-13.95 \pm 0.32$ | $2.65 \pm 0.03$ | $2.11 \pm 0.63$ |
| | | | | | MSE | | | | |
| | | carbon | concrete | energy | housing | kin8m | naval | power | yacht |
| ADAM | | $0.0025 \pm 0.0004$ | $24.23 \pm 2.47$ | $2.75 \pm 1.42$ | $23.89 \pm 1.27$ | $\mathbf{0.01 \pm 0.00}$ | $0.0022 \pm 0.0026$ | $11.24 \pm 0.39$ | $0.59 \pm 0.19$ |
| Pitfalls | $\beta = 0.5$ | $\mathbf{0.0023 \pm 0.0006}$ | $23.04 \pm 2.90$ | $0.69 \pm 0.14$ | $18.46 \pm 0.83$ | $\mathbf{0.01 \pm 0.00}$ | $\mathbf{0.0003 \pm 0.0002}$ | $10.70 \pm 0.50$ | $0.36 \pm 0.09$ |
| Pitfalls | $\beta = 1.0$ | $0.0028 \pm 0.0006$ | $23.00 \pm 3.16$ | $\mathbf{0.64 \pm 0.10}$ | $18.23 \pm 0.54$ | $\mathbf{0.01 \pm 0.00}$ | $\mathbf{0.0003 \pm 0.0001}$ | $\mathbf{10.61 \pm 0.54}$ | $0.47 \pm 0.12$ |
| Trustable-W2 | | $0.0029 \pm 0.0007$ | $\mathbf{21.93 \pm 1.67}$ | $0.68 \pm 0.15$ | $\mathbf{17.05 \pm 1.52}$ | $\mathbf{0.01 \pm 0.00}$ | $\mathbf{0.0003 \pm 0.0002}$ | $11.26 \pm 0.70$ | $\mathbf{0.33 \pm 0.11}$ |

