# OpenReview forum: "Stable Optimization of Gaussian Likelihoods"
_ICLR.cc/2023/Conference — Submitted to ICLR 2023_

### Official Review · Reviewer_gGLg · 2022-10-21

**Confidence:** 3
**Correctness:** 4
**Technical Novelty And Significance:** 3
**Empirical Novelty And Significance:** 3
**Recommendation:** 3

**Clarity, Quality, Novelty And Reproducibility:**

* The work is relatively clear. I have only noted a typo on p. 2: "according some Gaussian distribution". The limitations should be addressed more clearly.
* The analysis is sound and backed by empirical evidence.
* The authors did not include any supplementary material such as the code. However, reproducing the experiments should be doable.
* The novelty of the approach is somewhat not clear, as the work is a combination of previous research.  I am not familiar with some pieces of related research, such as TRPL. However, TRPL seems general enough to cover natural gradient updates? So the contribution of the authors seems to be in empirically studying the behavior of NG with trust regions.

**Strength And Weaknesses:**

*Strength*. The paper provides analysis of the source of the instability of the Gaussian NLL optimization. The authors did a multitude of experiments to provide evidence of the said instability. They provided necessary background information required to understand their approach. The method provides a more stable and in many cases better-performing optimization routine for Gaussian NLLs.

*Weakness*. The new optimization strategy is a combination of previous work. The new method seems to achieve superior results on synthetic data while on UCI datasets, results are quite close for different optimizers (table 7). Although Trustable does seem to produce a more stable descent (Figure 5).
Limitations of the method are not discussed clearly.

**Summary Of The Paper:**

The paper proposes an optimization method for the Gaussian likelihood. The authors argue that optimization of the Gaussian log-likelihood is prone to instabilities. The method is a combination of two approaches: using structured natural gradients with local parameterizations, and constraining them with trust region layers. The authors provide experimental evidence of the instabilities of the Gaussian NLL optimization and performance of their proposed method. The new method is more stable and performs on par or better than other methods such as ADAM or Pitfalls.

**Summary Of The Review:**

The paper provides a new strategy to optimize Gaussian NLL. The approach combines previously known methods. The authors should discuss more clearly the limitations of the method. My initial judgement is the paper is borderline.

EDIT: Given the lack of authors' response and in light of other reviews, I'd like to lower my score to 3: Reject.

---

### Official Review · Reviewer_YtD7 · 2022-10-24

**Confidence:** 3
**Correctness:** 4
**Technical Novelty And Significance:** 3
**Empirical Novelty And Significance:** 3
**Recommendation:** 3

**Clarity, Quality, Novelty And Reproducibility:**

Clarity & Quality: this paper is well-written.
Novelty: Novel
Reproducibility: The code was release on github.

**Details Of Ethics Concerns:**

The github link https://github.com/f6c334/iclr2023-stablenll gives some direct identity link to Denis Megerle, which might be a violation of anonymity.




**Strength And Weaknesses:**

Strengths:
1. Well motivated. This is indeed an critical problem for uncertainty learning.
2. Proposes good methedology from the perspective of eigenvalues of covariance and decoupled gradients. And also the good bulding of contextual gradients from the optimization process. All above brings the stability.
3. promising experimental results

Weakness:
1. Somewhat small scale experiment and also only regression is tested in the paper.

**Summary Of The Paper:**

This paper proposes Trustable, a novel optimizer that overcomes instabilities methodically by combining systematic update restrictions in the form of trust regions with structured, tractable natural gradients. It is demonstrated in several challenging experiments that Trustable outperforms current optimizers in regression with neural networks in terms of the NLL, MSE, and further performance metrics.

**Summary Of The Review:**

This paper is well-done except the experiments could be extended to larger-scale and more settings.The violation of anonymity is also found in the github repo.

---

### Official Review · Reviewer_WXLj · 2022-10-25

**Confidence:** 2
**Correctness:** 4
**Technical Novelty And Significance:** 3
**Empirical Novelty And Significance:** 2
**Recommendation:** 6

**Clarity, Quality, Novelty And Reproducibility:**

The paper is well written and the analysis was insightful. The algorithm is novel. I am not sure how easy it is to code up this algorithm and there seems to be no available implementation, as of now.

**Strength And Weaknesses:**

Pros:
A fairly novel method to solve the ensuing Gaussian likelihood optimization problem.
The experimental results do seem to indicate better performance than other competing methods.

Cons:
By very nature the methods seems to be unscalable (as it estimates a full covariance matrix) to higher dimension.s

**Summary Of The Paper:**

The authors consider the problem of Gaussian likelihood maximization. Estimating the conditional mean and the conditional covariance of a Gaussian likelihood model under neural modeling assumptions on the mean and the covariance leads to difficult optimization problems. The paper proposes a method to solve such optimization problem

**Summary Of The Review:**

Interesting algorithmic contribution, but not very scalable.

---

### Official Review · Reviewer_zKvQ · 2022-10-27

**Confidence:** 4
**Correctness:** 2
**Technical Novelty And Significance:** 2
**Empirical Novelty And Significance:** 2
**Recommendation:** 3

**Clarity, Quality, Novelty And Reproducibility:**

Clarity/Quality. I found this manuscript unclear and difficult to follow. Several sections are unnecessarily wordy and dense. Additionally, there are a few grammatical mistakes and quite a lot of odd choices of words or phrases that made the true underlying meaning ambiguous. Generally, the writing in this manuscript is of poor quality and could benefit from a few iterations of copy-editing. Lastly, the content is far from self-contained and requires familiarity with a number of existing works (Seitzer et al., 2022; Lin et al., 2021, Otto et al., 2021 to name the main ones) to understand the full extent of this paper's contribution.

Novelty. The novelty is somewhat limited. The main argument is that small eigenvalues in the covariance matrix cause the gradient to blow up, and conversely, large eigenvalues cause the gradient to saturate. This is somewhat obvious if not well-known. Furthermore, the proposed technique to mitigate the issues straightforwardly chains existing solutions together into a pipeline consisting of structured NG and TRPL.

Reproducibility. There is no mention of source code availability, nor any indication it will become available upon publication. I am not confident the results reported in this manuscript can be reproduced without it, due to the finicky nature of the constituent methods. Appendix A.3 contains hyperparameter settings and further training details which certainly helps, but I am not sure that this is sufficient.

**Strength And Weaknesses:**


## Strength

This paper seeks to tackle a timely and important problem, the solution for which would have a significant impact.

## Weakness

There is no strong theoretical/analytical justification for the proposed approach. It is well-known that small eigenvalues can cause the matrix to be non-PSD in practice due to the limitations of finite-precision floating-point arithmetic. We can usually reason about this precisely or construct failure examples by first assuming some unit round-off error u, constructing a PSD matrix with some properties in terms of u, and showing that finite-precision arithmetic can lead to small negative eigenvalues (and thus non-PSD-ness). For the proposed method to be convincing, one would ideally like to see some analysis and guarantees that it is robust or can recover, regardless of how large the round-off error is. This may be true by construction, but is not immediately obvious, and needs to be discussed in the manuscript. There is some discussion on Pg. 5, under "Structural Integrity", but the argument is a little hand-wavy. I am not entirely convinced that the proposed optimization method can actually always mitigate these issues. Instead, they may simply be less likely to encounter these problematic regions.

Concerning the experimental results, it appears the authors did tune the hyperparameters of their proposed approach on an intermediate validation/development set (according to A.3), but it is unclear whether the baselines against which they benchmarked their method benefited from any kind of similar tuning? Looking at Table 6, there appears to be a significant number of hyperparameters.

Second, I found the results themselves to be a bit underwhelming. In particular, the final test MSE often appears to be worse lower than the best-performing baseline? (e.g. Table 1-2)

Finally, there is no indication of performance in terms of runtime. Instead of merely comparing convergence in terms of the number of epochs (e.g. Figure 5), it is important to get a comparison in terms of at least one of wall-clock time, throughput (samples per sec), or FLOPs.

### Misc questions and issues

Here is a (non-exhaustive) list of some miscellaneous issues and questions:

Page 1:
- "fit only subpar"
Page 2:
- "empirically performing regions": meaning ambiguous
- "destructive gradients": meaning ambiguous
- "_Albeit_ all these efforts": "despite"
- "improved and less volatile fit in terms of [...]": vague
Page 3:
- "allow to"
Page 4:
- Figure 2 (a) "Univariant variance": univariate
Page 5:
- Figure 3 (a): this figure is quite difficult to parse. Also, there is nothing in the text dedicated to explaining what each item in the legend represents exactly. Most of these can be inferred but 'SqrtCovariance' -- does this just represent the matrix square root (Cholesky factorization)?
- "compensates large, but does not converge"
- "Vice versa"
- "rescaling is faulty"
- "potentially destructive changes"
- "disproportional": "disproportionate"
- "cannot sustain structure integrity": meaning ambiguous. Also, "structural integrity" is an odd choice of words.
- "requires to"
- "the covariance quickly drifts out of the parameter space of PSD matrices": please be more specific here. Presumably, you are referring to parameterizations that result, by construction, in a PSD covariance matrix such as the low-rank (Cholesky) or diagonal parameterization, and the reason they are not PSD is due to floating-point precision issues?
Page 6:
- "the gradient is _sensitised_ by"
- "degrading gradients further away": meaning ambiguous
- "we argue that these gradient explosions harm learning for neural networks": sentences like this are examples of where this manuscript is unnecessarily verbose. This "argument" can be unsaid because nobody would ever argue with this.
- "_inciting_ plateau regions"
- "parameter space integrity": the use of the word "integrity" here is non-standard and carries ambiguous meaning. I understand this to mean numerical stability and PSDness but this is not clear from the writing
- Normalization: I would use terms like "whitened", "standardized", or "de-correlated" here, because normalization could mean so many other things (e.g. projected onto unit hypersphere, affine transformed to be in unit hypercube, etc.)
Page 7:
- "_delivering_ favorable gradients"
- "cholesky": Upper case
- "respective the old distribution"
Page 8:
- "trade off the function approximator's _allocation_": meaning ambiguous - I understand you are trying to say its "representational capacity" or something along these lines but this is difficult to decipher.
Page 9:
- "without mean bound"
- "once converged, is stable": this sentence is tautological - if it's not stable, then it can't have converged?
- "multiple _windings_": ?
- "_especially_ the covariance cannot be fit"
- "but beta have a hard time fitting a full covariance"

**Summary Of The Paper:**

This paper focuses on the problem of optimising multivariate Gaussian likelihoods, specifically wrt to its mean and covariance matrix. They identify the root causes of various failure modes in Gaussian likelihood optimisation through the lens of spectral analysis. Specifically, they characterise these modes based on the covariance matrix's eigenvalues and propose combining numerous techniques into a pipeline to mitigate these issues. In particular, the methods consist of structured natural gradients (NG) and trust region projection layers (TRPL).

**Summary Of The Review:**

I recommend Rejection at this time. There are a number of significant issues in this paper that make it not yet ready for publication. In particular, as outlined above, there are gaps in the analysis of the proposed method and in the empirical results. Furthermore, there is room for improvement in the writing and presentation. That being said, this paper seeks to tackle an important problem, and could become a great contribution once the aforementioned issues are addressed.

EDIT: I am not inclined to update my score. The authors have not provided a response, and I have not found supportive arguments from other more positive reviews compelling enough to change my views.

---

### Decision · Program_Chairs · 2023-01-20

**Decision:**

Reject

**Justification For Why Not Higher Score:**

Clearly below the bar, see reviews. Also, there was no author feedback.

**Justification For Why Not Lower Score:**

N/A

**Metareview: Summary, Strengths And Weaknesses:**

This paper is concerned with optimizing mean and covariance of a multi-variate Gaussian distribution, which is a classical problem. There is substantial prior literature on this problem, in particular when it comes to estimating the conditional independence structure of the distribution, as revealed in the sparsity pattern of the inverse covariance. The authors point out that spectral properties of the covariance determine difficulty of optimization, and suggest some methodology.

Despite this being a basic statistical problem, with substantial amount of literature, no theoretical justifications are given for the proposed approach. Also, while it works well on synthetic data, it does not really outperform simpler baselines on more realistic problems.